# Separation and Semi-Empiric Modeling of Ethanol–Water Solutions by Pervaporation Using PDMS Membrane

**DOI:** 10.3390/polym13010093

**Published:** 2020-12-29

**Authors:** John Hervin Bermudez Jaimes, Mario Eusebio Torres Alvarez, Elenise Bannwart de Moraes, Maria Regina Wolf Maciel, Rubens Maciel Filho

**Affiliations:** School of Chemical Engineering, Separation Process Development Laboratory, State University of Campinas, Albert Einstein 500, Campinas 13083-582, Brazil; mariotame36@hotmail.com (M.E.T.A.); elenisebannwart@gmail.com (E.B.d.M.); wolf@unicamp.br (M.R.W.M.); rmaciel@unicamp.br (R.M.F.)

**Keywords:** pervaporation, Arrhenius, simulation, modeling, PDMS membrane, bioethanol

## Abstract

High energy demand, competitive fuel prices and the need for environmentally friendly processes have led to the constant development of the alcohol industry. Pervaporation is seen as a separation process, with low energy consumption, which has a high potential for application in the fermentation and dehydration of ethanol. This work presents the experimental ethanol recovery by pervaporation and the semi-empirical model of partial fluxes. Total permeate fluxes between 15.6–68.6 mol m${}^{-2}$ h${}^{-1}$ (289–1565 g m${}^{-2}$ h${}^{-1}$), separation factor between 3.4–6.4 and ethanol molar fraction between 16–171 mM (4–35 wt%) were obtained using ethanol feed concentrations between 4–37 mM (1–9 wt%), temperature between 34–50 ${}^{\circ }$C and commercial polydimethylsiloxane (PDMS) membrane. From the experimental data a semi-empirical model describing the behavior of partial-permeate fluxes was developed considering the effect of both the temperature and the composition of the feed, and the behavior of the apparent activation energy. Therefore, the model obtained shows a modified Arrhenius-type behavior that calculates with high precision the partial-permeate fluxes. Furthermore, the versatility of the model was demonstrated in process such as ethanol recovery and both ethanol and butanol dehydration.

## 1. Introduction

Pervaporation is a membrane separation process used in the separations of mixtures, such as water–organic [1], organic–water [2] or organic–organic [3]. In the pervaporation separation, a membrane acts as the separating barrier for the component of minor affinity. When both the membrane and feed are in contact, some molecules can be recovered from the feed due to its higher affinity and quicker diffusivity in the membrane [4], which can be carried out applying a differential pressure between the membrane walls through a vacuum pump or a carrier gas [5]. The main advantage of pervaporation is the low energy consumption compared with traditional processes such as distillation and liquid-liquid extraction [6,7,8], but also the possibility to work at moderate temperature can be an advantage for the separation of temperature sensitive products, be an environmentally friendly process [9], reduces the cost of production, generates products free from solvent contamination and can be adapted to both continuous and batch processes [10]. Initially, pervaporation was intended for the selective separation of azeotropic mixtures. Currently, its application extends to various areas of industry, standing out in the extraction of aromas (alcohols, esters, organic compounds) from agro-food systems (wastes, by-products, fruit juices, food processed products), ethanol removal from alcoholic drinks towards the production of non-alcoholic beverages [11], development of chemical (water removal: esterification, acetalization, ketalization, etherification) and bio-chemical reactions (alcohol production) [12,13,14,15,16,17,18], dehydration of organics (methanol, ethanol, isopropanol, butanol) [19,20,21,22] and waste water treatment [23]. Pervaporation can be coupled to fermentation creating a hybrid pervaporation-fermentation process, which is used to recovery the bioproduct, such as acetone-butanol-ethanol (ABE) [24], butanol [25], and ethanol [26], in order to eliminate inhibition products, further improvement on product productivity and enhancement of the substrate conversion rate [27]. In the alcohol industry, pervaporation is gaining space in hybrid systems, such as distillation–pervaporation and fermentation–pervaporation, in which the pervaporation membrane can be located inside the main unit or in an external pervaporation module [28,29,30,31]. Therefore, the development of pervaporation models are essentials in the study of these systems.

Mass transfer for the separation of binary solutions in pervaporation can be described using semi-empirical models. The main models are developed from the mathematical description of two independent variables, measured experimentally, such as total permeate flux and separation factor [32], permeability of permeants [33] or permeate fluxes [34] and the model parameters are mainly calculated by objective functions [35,36]. Most of the reported models are based on the solution–diffusion model [37]. However, it is common to use the Arrhenius model to describe the dependence of permeate fluxes on temperature [38]. In general, pervaporation can be described using the solution–diffusion model [39] and the Arrhenius equation or a combination of them, which are used to describe the permeate flux mainly. However, the solution–diffusion model is the most accepted to describe the transport of mass across the membrane [40]. The solution–diffusion model involves a more complex mathematical development than that required by the Arrhenius model. In this sense, the latter ones have gained space, being used in the modeling of processes for industrial application [41]. Table 1 shows some permeate flux models reported in the literature.

Commonly, pervaporation models are based on the mathematical modeling of permeate flux (*J*) and the separation factor ($\beta_{ij}$). However, to compare the performance of the membranes it is necessary to present the results based on driving force normalized properties such as permeability ($\prod_{i}$), permeance ($\prod_{i}/\ell$) or membrane selectivity ($\alpha_{ij}$) as proposed by [42].

The aim of this work was to develop a semi-empirical model from the analysis of the effects of temperature and feed concentration on the partial-permeate flux (ethanol and water), using a commercial polydimethylsiloxane (PDMS) membrane. The relative mathematical simplicity of this semi-empirical model, its high adjusted R-squared and low mean square error promote its application in the study of industrial processes such as fermentation–pervaporation or distillation–pervaporation.

**Table 1 polymers-13-00093-t001:** Permeate flux models.

Model	Ref.
$J_{i}=\frac{1}{1+\bar{D_{i}}\exp\left(B\;\cdot \;x_{i1}\right)/\left(Q_{0}\;\cdot \;P_{i0}\;\cdot \;\bar{\gamma_{i}}\right)}\;\cdot \;\frac{\bar{D_{i}}\exp\left(B\;\cdot \;x_{i1}\right)}{\bar{\gamma_{i}}}\;\cdot \;\left(\frac{P_{i1}-P_{13}}{P_{i0}}\right)$	[36]
$\bar{D_{i}}=\bar{D_{i}^{*}}\exp\left[\frac{E_{i}}{R}\left(\frac{1}{T^{*}}-\frac{1}{T}\right)\right]$; $\bar{\gamma_{i}}=\sqrt{\gamma_{i1}\gamma_{i3}}$; $OF=\sum_{1}^{n}{\left(\frac{J_{i,mesured}-J_{i,\mathrm{modeled}}}{J_{i,mesured}}\right)}^{2}$	
$J_{i}=ax_{i}^{2}+bx_{i}+c$	[43]
a,b,c: parameters	
$J_{i}=\frac{D_{ij}}{L}\left(Con_{i}^{f}-Con_{i}^{p}\right)$	[44]
$D_{ij}=D_{0}e^{\left(\frac{E_{0}}{T+273.15}\right)}$; $Con_{i}^{p}=\frac{P^{p}}{RT^{p}{\left(1+\frac{B}{V_{m}}\right)}^{y_{i}^{p}}}$	
$J_{i}=\omega_{i}\exp\left(\varepsilon_{i}x_{i}\right)\exp\left[\frac{E_{i}}{R}\left(\frac{1}{T_{ref}}-\frac{1}{T}\right)\right]\left(x_{i}\gamma_{i}P_{i}^{sat}-y_{i}P_{p}\right)$	[45]
$\omega_{i},\varepsilon_{i},E_{i}$: parameters	
$J_{i}=L_{i}\left(T,x_{i}\right)\left[P_{i}^{0}\left(T_{F}\right)x_{F,i}\gamma_{F,i}-P_{p}y_{p,i}\right]$	[46]
$L_{i}\left(T,x_{i}\right)=L_{0,i}\left(T_{0},x_{i}\right)\exp\left[-\frac{E_{Act,i}\left(x_{i}\right)}{R}\left(\frac{1}{T_{0}}-\frac{1}{T_{f}}\right)\right]$	
$L_{i}\left(T,x_{H_{2}O}\right)=a_{i}\exp\left[b_{i}\left(\frac{1}{T_{0}}-\frac{1}{T_{f}}\right)+c_{i}x_{H_{2}O}\right]$ (hydrophilic membrane)	
$L_{i}\left(T,x_{BuOH}\right)=\left(a_{i}\ln\left(x_{BuOH}\right)+b_{i}\right)\exp\left[c_{i}\left(\frac{1}{T_{0}}-\frac{1}{T_{f}}\right)\right]$ (hydrophobic membrane)	
$J_{i}=\frac{D_{i}^{0}}{L}\left(\frac{c_{i}^{2}}{2}+\beta_{ij}c_{i}\left(1-c_{i}\right)\right)$	[47]
$D_{i}^{0},\beta_{ij}$: parameters	
$J_{W}=\left(P_{Wsat}^{F}\gamma_{W}^{F}x_{W}^{F}-P_{W}^{P}\right)\;\cdot \;K_{1}\exp\left(\frac{K_{2}}{T}\right)$	[48]
$J_{P}=\left(P_{Psat}^{F}\gamma_{P}^{F}x_{P}^{F}-P_{P}^{P}\right)\exp\left(K_{3}+\frac{K_{4}}{T}\right)\left[1+J_{W}\exp\left(K_{5}+\frac{K_{6}}{T}\right)\right]$	
$K_{1-6}$: parameters	
$J_{i}={\left.\frac{DC^{m}}{l\gamma^{m}}\right|}_{i}\left(\gamma_{l,i}x_{l,i}-\frac{P_{2}x_{2,i}}{P_{v,i}}\exp\left\{\frac{v_{i}\left(P_{v,i}-P_{l}\right)}{RT}\right\}\right)$	[49]
$J=A_{0}\exp\left(\frac{-E_{p}}{T}\right)$	[50]
$y={\left(\left(\left(440.9-\frac{112700}{T}\right)x\right){}^{2}+98.3^{-2}\right)}^{-0.5}$	
$A_{0}=2995\exp\left(2.8441x\right)$; $E_{p}=330.04+839.58x$; $a=440.9-\frac{112700}{T}$	
$J_{1}={\bar{D}}_{1,m}\left(\frac{w_{1}{\bar{D}}_{2,m}+D_{12}}{D_{12}+w_{1}{\bar{D}}_{2,m}+w_{2}{\bar{D}}_{1,m}}\right)\rho_{m}\frac{\Delta w_{1}}{\delta }+{\bar{D}}_{1,m}\left(\frac{w_{1}{\bar{D}}_{2,m}}{D_{12}+w_{1}{\bar{D}}_{2,m}+w_{2}{\bar{D}}_{1,m}}\right)\rho_{m}\frac{\Delta w_{2}}{\delta }$	[51]
$J_{2}={\bar{D}}_{2,m}\left(\frac{w_{2}{\bar{D}}_{1,m}+D_{12}}{D_{12}+w_{2}{\bar{D}}_{1,m}+w_{1}{\bar{D}}_{2,m}}\right)\rho_{m}\frac{\Delta w_{2}}{\delta }+{\bar{D}}_{2,m}\left(\frac{w_{2}{\bar{D}}_{1,m}}{D_{12}+w_{2}{\bar{D}}_{1,m}+w_{1}{\bar{D}}_{2,m}}\right)\rho_{m}\frac{\Delta w_{1}}{\delta }$	
${\bar{D}}_{1,m}=\frac{D_{10}\left[\exp\left(\varepsilon_{11}{\bar{w}}_{1}+\varepsilon_{12}w_{2,F}\right)-\exp\left(\varepsilon_{11}{\bar{w}}_{1}+\varepsilon_{12}w_{2,P}\right)\right]}{\varepsilon_{12}\left(w_{2,F}-w_{2,P}\right)}$	
${\bar{D}}_{2,m}=\frac{D_{20}\left[\exp\left(\varepsilon_{21}{\bar{w}}_{1}+\varepsilon_{22}w_{2,F}\right)-\exp\left(\varepsilon_{21}{\bar{w}}_{1}+\varepsilon_{22}w_{2,P}\right)\right]}{\varepsilon_{22}\left(w_{2,F}-w_{2,P}\right)}$	

## 2. Materials and Methods

### 2.1. Reagent

Absolute ethanol from JT Baker.

### 2.2. Ethanol Quantification

Ethanol permeate was determined by high-performance liquid chromatography (HPLC), employing a chromatograph (Agilent 1260, Campinas, SP, Brazil) equipped with a refractive index detector and a Bio-Rad Aminex HPX-87H column (300 × 7.8
mm) operated at 30 ${}^{\circ }C$ and sample injection volume of 20 μL. The eluent used was 5 mM H_2_SO_4_, at a flow rate of 0.6
mL
$\min^{-1}$. Solutions of ethanol between 0.1–4.8 wt% were used as standards [52].

### 2.3. Equipment

Pervaporation tests were conducted using a bioreactor of 5 L (model BioFlo and CelliGen 310, New Brunswick Scientific, Campinas, SP, Brazil), peristaltic pump (model 620s, Watson-Marlow, Campinas, SP, Brazil), coupled with a pervaporation system developed by the author, composed by temperature sensor (pt 100), temperature controller (model N1040, Novus, Canoas, RS, Brasil), digital vacuum gauge (Cole-Parmer, Campinas, SP, Brazil), Dimroth condenser jacketed, thermostatic bath (Marconi, Piracicaba, SP, Brazil), vacuum pump (model RV8, Labconco, Campinas, SP, Brazil) and a commercial polydimethylsiloxane (PDMS) tubular membrane (organophilic PDMS membrane onto ceramic carrier tube, dimensions (out × in): 10 × 7 mm, tube: 25 cm, active area: 48 c$m^{2}$, thickness: 3–5 μm). Figure 1 shows the whole experimental pervaporation system.

### 2.4. Experimental Test

At the start of each test, the pervaporation unit was stabilized for 30 min circulating the alcoholic solution between the vessel and the membrane at 280 m$L^{-1}$/h that is sufficient to maintain the flow in a turbulent region in all experimental conditions (Reynolds number higher than 13,000) and keeping both the ethanol concentration (4, 12, 20, 29 and 37mM equivalent to 1, 3, 5, 7 and 9 wt%, respectively) and the temperature (34, 40, 45 and 50 ${}^{\circ }C$) at the study conditions. At the same time, the condenser was stabilized at 12 mbar and −6
${}^{\circ }C$. Subsequently, the valve located between the membrane and the vacuum gauge was opened and the permeate was condensed and collected for 1.5
h. The condensed permeate was weighed and the ethanol was quantified by HPLC. Permeate fluxes (ethanol ($J_{i}$), water ($J_{j}$) and total ($J_{t}$)) and separation factor ($\beta_{ij}$) were calculated according to Equations (Equation 1)–(Equation 4), respectively.
(1)$$J_{i}=\frac{m_{tot}}{A\times t}\times \frac{\mathrm{wt}{\%}_{i}}{100}$$
(2)$$J_{j}=\frac{m_{tot}}{A\times t}\times \frac{100-\mathrm{wt}{\%}_{i}}{100}$$
(3)$$J_{t}=J_{i}+J_{j}$$
(4)$$\beta_{ij}=\frac{{\left(Con_{i}/Con_{j}\right)}^{p}}{{\left(Con_{i}/Con_{j}\right)}^{f}}$$

Permeability ($\prod_{i}$, 1 Barrer ≈ 1.20546 × 10 ^−12^ mol m^−1^ h^−1^ Pa^−1^) and membrane selectivity were calculated by Equations (Equation 5) and (Equation 6), respectively. The saturated vapor pressure ($P_{i}^{sat}$) and activity coefficients ($\gamma_{i}$) for ethanol and water were determined using the Extended Antoine equation and the Non-Random Two-Liquid (NRTL) model, obtaining the coefficients from ASPEN Plus V11.
(5)$$\prod {}_{i}=\frac{\left(J_{i}/Mw_{i}\right)\ell }{x_{i}\gamma_{i}P_{i}^{sat}-y_{i}P^{p}}$$
(6)$$\alpha_{ij}=\frac{\prod {}_{i}}{\prod {}_{j}}$$

### 2.5. Semi-Empirical Model for Flux Determination

The semi-empirical method for flux determination was developed by the analysis of the behavior of the permeate flux, permeate concentration and permeation temperature and the model parameters obtained using OriginLab software, which calculates the model parameters internally using the method of Partial Least Squares (PLS) [53,54]. Then, the observed mathematical model was added to the software surface database. Subsequently, the experimental data (permeate concentration, temperature and partial flux) were plotted using the Nonlinear Surface Fit option and the model parameters were calculated.

The performance of the model obtained was evaluated in terms of the adjusted R-squared ($adj-R^{2}$), Equation (Equation 7), [55] and the root mean square error (RMSE), Equation (Equation 8), [56].
(7)$$adj-R^{2}=1-\frac{\sum_{i=1}^{n}{\left(J_{\exp,i}-J_{i}\right)}^{2}/df_{err}}{\sum_{i=1}^{n}{\left(J_{\exp,i}-{\bar{J}}_{\exp}\right)}^{2}/df_{tot}}$$
(8)$$RMSE\left(\%\right)=1-\frac{\sqrt{\frac{\sum_{i=1}^{n}{\left(J_{\exp,i}-J_{i}\right)}^{2}}{n}}}{{\bar{J}}_{\exp}}\times 100$$

The model fit was verified both with the experimental data presented in this work and by its application in the separation of binary components by pervaporation reported in the literature. The literature data were extracted from the flux behavior graphs using digitize image tool of the OriginLab software. Later, the model parameters were obtained and the evaluation criteria ($adj-R^{2}$ and RMSE) were calculated.

## 3. Results

### 3.1. Pervaporation Performance

Ethanol and water separation performance through PDMS commercial membrane was evaluated based on feed composition and temperature. Under operating condition, total permeate flux between 15.6–68.6 mol m${}^{-2}$ h${}^{-1}$ (289–1565 g m${}^{-2}$ h${}^{-1}$) and separation factor between 3.4–6.4 were obtained (Figure 2).

The results are consistent with the characteristics of the membrane reported by the provider (500–1000 g m${}^{-2}$ h${}^{-1}$, separation factor of 6; ethanol 5 v%, 5–10 mbar) and are consistent with the literature [57]. As observed in Figure 2a, total permeate flux increased with both ethanol concentration and temperature, while the separation factor showed a slight drop with increasing of ethanol feed concentration and increased with temperature. The observed total flux performance and the decrease in separation factor in PDMS membrane are well known [58]. However, the increases in the separation factor with the increase in feed temperature is a phenomenon that is not always observed (Figure 2b). On some occasions the separation factor tends to decrease [59] as a consequence of the loss of the hydrophobic character of the membrane, largely attributed to membrane swelling. Studies carried out by Wang et al. [60] show that the temperature can affect in a lesser degree, the swelling of PDMS membrane when compared to the effect produced by the ethanol feed concentration. It was observed small increases in the separation factor with increasing temperature and decrease of separation factor with increasing of ethanol feed concentration.

Figure 3a shows the permeability profile of the membrane under working conditions, observing permeabilities between 4205–5618 Barrer for ethanol and between 8205–14,787 Barrer which are in the estimated range for PDMS membranes (Table 2).

In general, the ethanol concentration showed a slight increase in the permeability of the permeants, effect that increased with the increase in temperature. However, it is observed that the permeability of water is higher than that of ethanol, obtaining selectivity between 0.3–0.5 (Figure 3b) with a favorable temperature effect. Therefore, this commercial membrane did not show ethanol selectivity. In fact, studies carried out by Rozcika et al. 2014 [58] showed that the commercial membranes Pervap 4060, Pervatech and PolyAn do not show ethanol selectivity, very possible due to the membrane preparation method [61] or the chemical composition of the active membrane layer. Although it is known that PDMS is benchmark material in the preparation of ethanol perm-selective membranes, pure PDMS membranes show low fluxes and little ethanol selectivity. However, by modifying the composition of the PDMS membrane (mixed-matrix membranes (MMMs) or hybrid membranes) it is possible to improve the ethanol separation efficiency, increasing both permeability and selectivity [59,62,63,64].

**Table 2 polymers-13-00093-t002:** Comparison of various PDMS membrane performances in the ethanol–water mixture separation.

Feed		Pervaporation				Membrane		Ref.
wt%${}_{i}$	T (${}^{\circ }$C)	$p_{\mathrm{ethanol}}$ (Barrer)	$p_{\mathrm{water}}$ (Barrer)	$\alpha_{\mathrm{ij}}$	*P* (Pa)	Composition	*ℓ* (μm)	
5	25	*—*	*—*	0.7	*—*	Pervap 4060 ${}^{a}$	*—*	[58]
5	25	*—*	*—*	0.6	*—*	Pervatech ${}^{a}$	*—*	[58]
5	25	*—*	*—*	0.6	*—*	PolyAn ${}^{a}$	*—*	[58]
10–25	40–60	7210–8345	9043–11,292	0.6–0.9	500	PDMS	30	[59]
10–25	40–60	32,294–43,743	20,883–64,829	0.7–1.9	500	POSS-g-PDMS	30	[59]
2	60	10,368	17,034	0.6	*—*	PDMS	*—*	[62]
2	60	17,914	23,315	0.8	*—*	ZIF-71/PDMS	*—*	[62]
6	37–69	12,555–16,920	17,159–20,564	0.7–0.9	300	PDMS	9	[63]
2–10	37–69	22,899–34,756	29,212–39,639	0.6–1.0	300	PDVB-coated PDMS	15	[63]
10–50	40–60	17,096–30,601	12,734–17,758	1.2–1.7	200	OPS/PDMS	30	[64]
3–16	20–40	*—*	*—*	0.9–1.0	50	PDMS	*—*	[65]
1–10	40–70	5594–16,024	4559–9202	1.0–1.8	300	PDMS/ZIF-8	1.16	[66]
5	50	49,873	25,564	2.0	532	PDMS vinyl	83	[67]
19	34–50	4145–9252	12,368–17,259	0.3–0.5	1200	PDMS ${}^{a}$	4	This work

^*a*^ Commercial membrane.

### 3.2. Effect of Feed Concentration

Figure 4 shows the effect of ethanol feed concentration on the ethanol (Figure 4a) and water (Figure 4b) fluxes at different temperatures. Under these conditions, ethanol flux increased proportionally with increasing of ethanol feed concentration while the water flux presented a slight increase, as reported in the literature [68]. Studies have shown that increasing the ethanol feed concentration normally result in the increase of swelling degree of membrane, free volume of membrane [60], partial pressure [69] and improves the affinity of ethanol to membrane [70]; increasing the driving force and consequently the ethanol flux in permeate. Moreover, the small diameter of water molecules facilitates its transport through the free volume of membrane observing higher water fluxes with approximately constant behaviors [71]. Furthermore, the slight increase in the water flux permeate led to a decrease in the separation factor (Figure 2b), as reported in the literature [72].

### 3.3. Effect of Feed Temperature

Figure 5a shows the effect of feed temperature on both ethanol and water fluxes permeate at different ethanol feed concentration.

As it can be seen, for all the ethanol concentrations studied (1–9 wt%), the ethanol and water fluxes increased exponentially with increasing of feed temperature, which is considered to be Arrhenius behavior, Equation (Equation 9), [63].
(9)$$J_{i}\left(T\right)=J_{0,i}\exp\left[\frac{-Ea_{i}}{R\left(T+273.15\right)}\right]$$

The effect of temperature on permeate is complex. Increasing the temperature, the kinetic energy of the feed molecules increases and in contact with the membrane, increases the mobility of the PDMS chain and the free volume, in addition to increasing the saturated vapor pressure (greater increase for ethanol). This leads to an increase in the transport of mass through the membrane [73] and consequently an increase in permeate fluxes. However, the composition of the membrane can modify the solubility of the permeants and consequently affects the separation factor. According to Figure 5a it is expected that the increase in temperature will increase the kinetic energy of the permeants [16], permeate fluxes, the vapor pressure [74], the swelling membrane [75] and the free volume in the membrane [76]. Despite the increase in water permeate flux, the solubility towards ethanol in the membrane surface was favorable, the ethanol permeate concentration increased slightly (Figure 5b) and consequently a small increase in the separation factor, such as observed in Figure 2b and reported in the literature [66].

Considering the Arrhenius equation behavior of the permeate fluxes, the apparent activation energy (Ea) was calculated from the slope of LnJ vs. 1/T [73], for each ethanol feed concentrations evaluated (1, 3, 5, 7 and 9 wt%). Ethanol molecules exhibited higher apparent activation energy than those of water, 82.7–84.7 kJ mol${}^{-1}$ and 59.5–62.2 kJ mol${}^{-1}$, respectively. This indicates that permeation rate of ethanol molecules is more sensitive to the feed temperature and, consequently, the separation factor increases under these conditions [16]. Likewise, a relationship was observed between apparent activity energy and ethanol feed concentration (Figure 6).

This behavior is very little reported in the literature, because the authors only consider a single feed concentration in their research. However, Zhou et al. [69] have reported the dependence of the apparent activation energy with the feed concentration in binary solutions of acetone/water, butanol/water and ethanol/water using a silicate/PDMS membrane. According to the results obtained in this study, it is possible to observe a linear trend for the apparent activation energies of permeate, as reported by Yeom et al. [77].

### 3.4. Semi-Empirical Model

The model developed in this study is based on the behavior analysis of the partial-permeate flux (ethanol and water) under the conditions of both ethanol concentration and temperature in the feed. As observed in this study, from the Arrhenius equation, the behavior of permeate fluxes can be described as a function of temperature, Equation (Equation 9). However, Figure 4 shows that there is a directly proportional relationship between the permeate fluxes and the concentration of the permeant in the feed. However, it must be taken into account that this phenomenon is observed when the concentration of the permeating component is low. Therefore, in this specific case, it is only observed for ethanol. This relationship will be expressed by Equation (Equation 10).
(10)$$J_{0,i}\left(Con_{i}^{f}\right)=aCon_{i}^{f}$$
In this way, the permeate fluxes are expressed by Equation (Equation 11).
(11)$$J_{i}\left(Con_{i}^{f},T\right)=aCon_{i}^{f}\exp\left[\frac{-Ea_{i}}{R\left(T+273.15\right)}\right]$$
Likewise, Figure 6 shows that the apparent activation energy is sensitive to changes in the concentration of permeants, for which a linear behavior is suggested, Equation (Equation 12).
(12)$$-Ea_{i}\left(Con_{i}^{f}\right)=bCon_{i}^{f}+c$$
Finally, from these observations, the behavior of partial-permeate fluxes is mathematically expressed from the temperature and of feed concentration, Equation (Equation 13). It must be taken into account that the concentration of the species (*i* and *j*) in the feed are not kinetic, for example this may be related to the effects of concentration polarization.
(13)$$J_{i}\left(Con_{i}^{f},T\right)=aCon_{i}^{f}\exp\left[\frac{bCon_{i}^{f}+c}{R\left(T+273.15\right)}\right]$$

To have a better mathematical description of separation, it is necessary to reduce effects that are not part of the phenomenon, for example unexpected changes in the partial fluxes due to fluctuations in the permeate pressure. Parameters *a*, *b* and *c* of the model, Equation (Equation 13), can be obtained by fitting the experimental data of the partial-permeate fluxes by method of least squares [78]. In this study, model parameters were obtained using OriginLab software (Table 3). Parameter *a* represents the sensitivity of the Arrhenius coefficient ($J_{0}$) to changes in the concentration of the permeate component in the feed; parameter *b* represents the sensitivity of the apparent activation energy to changes in the feed concentration and parameter *c* indicates the apparent activation energy.

Figure 7 shows the experimental data of the partial-permeate fluxes and those calculated by the model. The proximity between the experimental and calculated fluxes indicates that the model predicts the behavior of the fluxes with high accuracy. Adjusted R-squared ($Adj-R^{2}$) values close to 1 (0.9956 and 0.9967) confirm the good fit of the model for both ethanol and water fluxes, respectively, while the low RMSE values confirm the high accuracy of the model (5.45 and 2.33% for ethanol and water, respectively). According to Li et al. [56], model accuracy is considered excellent when RMSE<10%, good if 10%<RMSE<20%, fair if 20%<RMSE<30%, and poor if RMSE>30%. From this point of view, the developed model calculates with excellent precision the partial-permeate fluxes.

### 3.5. Partial Flux Model Application

Considering the experimental data reported in the literature, the versatility of the model was verified for recovering and dehydration of alcohols at different pervaporation conditions (Table 4).

As observed in Table 5, the model presented good fit ($Adj-R^{2}$ close to one) and excellent accuracy (RMSE lesser than 10%), when evaluated for ethanol recovering and dehydration of both ethanol and butanol by pervaporation (Appendix A).

In principle, the high degree of fit of the experimental data of the model is due to the fact that its development was based, in the first place, on the effect of feed temperature on the partial-permeate flux, which is characterized by presenting the behavior of Arrhenius. For decades, this behavior has been widely known and accepted by researchers. The addition of the effect produced by the feed concentration and the dependence of the apparent activation energy on the feed concentration adjust the Arrhenius equation to the separation characteristics of the membrane; providing high adjusted R-squared and low mean square error. It is important to highlight that of the literature examples presented on ethanol recovery, the membranes used by Zhan et al. 2020 [59] and Mao et al. 2019 [66] presented ethanol selectivity, demonstrating the efficiency of the model.

## 4. Conclusions

According to the results, semi-empirical model was developed to predict the performance of the partial-permeate fluxes as a function of the temperature and feed concentration with a good fit of the experimental data and very good accuracy.

The effect of the temperature and the feed concentration on the permeate flux led to the adjustment of the Arrhenius equation by modifying the Arrhenius coefficient and the apparent activation energy; modifications that allow the Arrhenius equation to be adapted to the separation characteristics of the membrane. From the models reported in the literature, the model developed presents similarity with the reported by Yeom et al. (2020) [50] which differs in the calculation of the pre-exponential factor.

The developed model can be characterized by presenting a relatively simple mathematical equation and versatile. Its mathematical model facilitates the calculation of the model parameters and their application; and its versatility allows it to be used to predict the partial-permeate fluxes of hydrophilic, hydrophobic and even organophilic pervaporation in single or hybrid process.

Although the membrane used in the development of the model did not show ethanol selectivity, the model developed showed a high degree of fit with the data obtained from Mao et al. 2019 [66] and Zhan et al. 2020 [59] who used selective membranes.

This model is adapted to processes in which the partial fluxes is proportional to the feed concentration at low concentration and exponential with feed temperature. In the case that the effect of a single variable (concentration or temperature) is studied, the other variable can be set, and the number of experiments decreased. However, as it is a data-fitting model, it is recommended that the experimental data show defined trends with little deviation. If this is the case, the tests showing significant process deviations must be repeated.

## Figures and Tables

**Figure 1 polymers-13-00093-f001:**
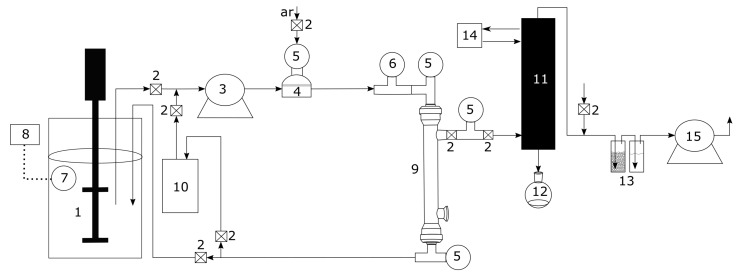
Schematic of the pervaporation experimental setup used in the study. (1) vessel, (2) valve, (3) peristaltic pump, (4) pulse damper, (5) pressure gauge, (6) vacuum gauge, (7) temperature sensor, (8) temperature controller, (9) membrane and module, (10) vessel, (11) Jacket Dimroth condenser, (12) product trap, (13) safety trap, (14) thermostatic bath, (15) vacuum pump.

**Figure 2 polymers-13-00093-f002:**
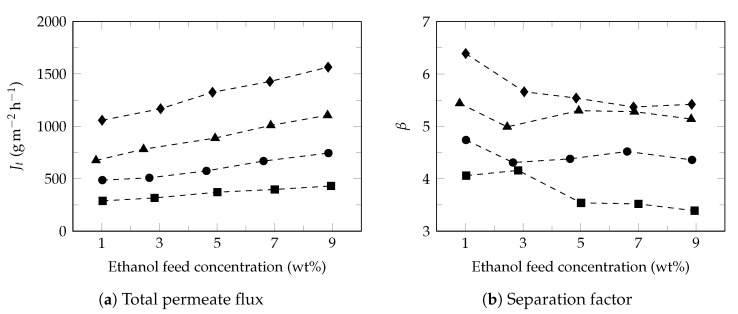
Total flux and separation factor. Feed temperature: 34 ${}^{\circ }C$ (⯀), 40 ${}^{\circ }C$ (●), 45 ${}^{\circ }C$ (▲) and 50 ${}^{\circ }C$ (♦).

**Figure 3 polymers-13-00093-f003:**
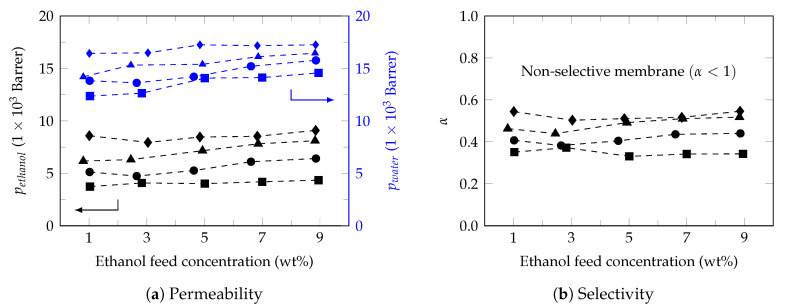
Permeability and membrane selectivity. Feed temperature: 34 ${}^{\circ }C$ (⯀), 40 ${}^{\circ }C$ (●), 45 ${}^{\circ }C$ (▲) and 50 ${}^{\circ }C$ (♦).

**Figure 4 polymers-13-00093-f004:**
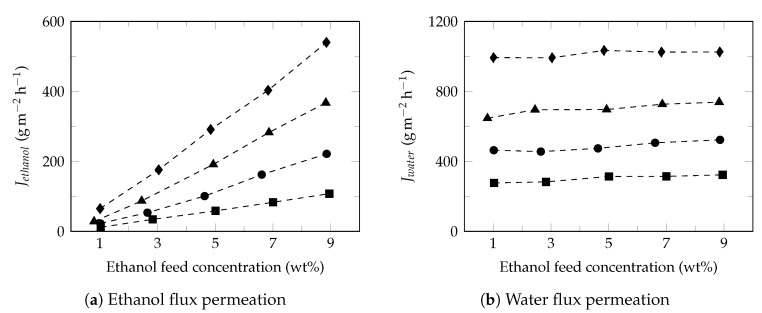
Effect of ethanol feed concentration on the ethanol and water fluxes. Feed temperature: 34 ${}^{\circ }C$ (⯀), 40 ${}^{\circ }C$ (●), 45 ${}^{\circ }C$ (▲) and 50 ${}^{\circ }C$ (♦).

**Figure 5 polymers-13-00093-f005:**
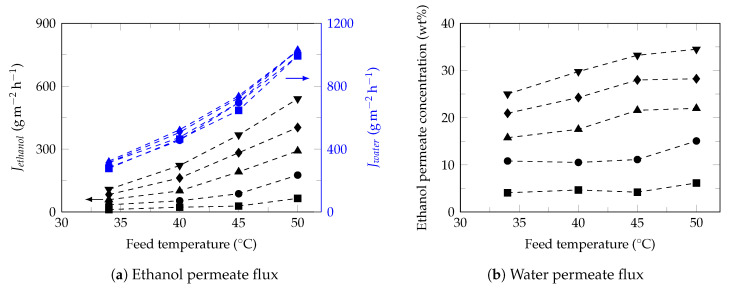
Effect of feed temperature on the ethanol (black) and water (blue) fluxes. Ethanol feed concentrations: 1 wt% (⯀), 3 wt% (●), 5 wt% (▲), 7 wt% (♦) and 9 wt% (▼).

**Figure 6 polymers-13-00093-f006:**
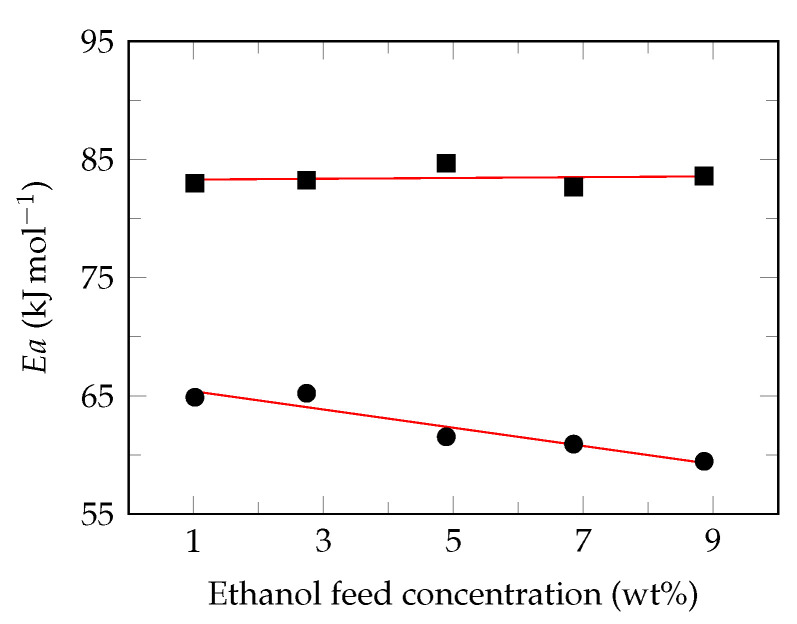
Effect of Ethanol feed concentration on the apparent activation energy. Ethanol (⯀), water (●), suggested behavior (—).

**Figure 7 polymers-13-00093-f007:**
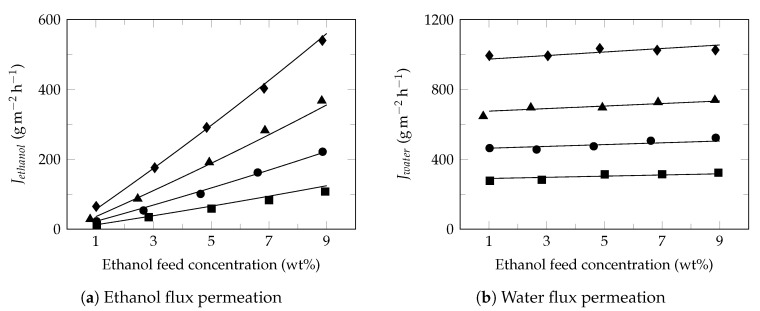
Ethanol and water fluxes experimental (symbol) and simulation (line). Feed temperature: 34 ${}^{\circ }C$ (⯀), 40 ${}^{\circ }C$ (●), 45 ${}^{\circ }C$ (▲) and 50 ${}^{\circ }C$ (♦).

**Table 3 polymers-13-00093-t003:** Calculated model parameters.

Parameter	Ethanol	Water	Unit
*a*	2.1847 × 10^14^	1.2170 × 10^11^	g m^−2^ h^−1^
*b*	3.0433 × 10^1^	−5.4992 × 10^1^	J mol^−1^
*c*	−7.7889 × 10^4^	−5.6995 × 10^4^	J mol^−1^

**Table 4 polymers-13-00093-t004:** Experimental pervaporation conditions.

Feed		Pervaporation			Membrane			Ref.
wt%${}_{i}$	*T* (${}^{\circ }$C)	$J_{t}$ (mol m${}^{-2}$ h${}^{-1}$)	$\beta_{\mathrm{ij}}$	*P* (Pa)	Composition	*A* (cm${}^{2}$)	*ℓ* (μm)	
1–20	30–50	2673–9704	4.8–5.4	150	PDMS/UiO-66-TMS ${}^{b}$	19.6	*—*	[60]
10–25	40–60	293–1792	3.7–17.9	500	POSS-g-PDMS ${}^{b}$	22	30	[59]
1–10	40–70	1387–4417	9.7–20.6	300	PDMS/ZIF-8 ${}^{b}$	20	1.16	[66]
5–20	25–60	982–4448	3.0–5.6	300	MAF-6/PEBA ${}^{b}$	*—*	5	[79]
3–11	25–55	3410–21,147	5.9–7.0	*—*	PDMDES ${}^{b}$	55.4	1	[80]
20–80	70–90	15,335–98,933	15.7–101.9	500	PVA ${}^{c}$	28.3	20	[81]
5.1–15.8	60–100	685–4819	32.8–188.8	300	Pervap 2510 ${}^{a,d}$	178	*—*	[45]
1–9	3–50	288–1565	3.4–6.4	1200	PDMS ${}^{a,b}$	50	4	This work

^*a*^ commercial membrane, ^*b*^ ethanol recovery, ^*c*^ ethanol dehydration, ^*d*^ butanol dehydration.

**Table 5 polymers-13-00093-t005:** Partial flux model application on experimental pervaporation

Component		*a*	*b*	*c*	$Adj-R^{2}$	*RMSE*	Ref.
${}_{ij}$		g m${}^{-2}$ h${}^{-1}$	J mol${}^{-1}$	J mol${}^{-1}$		%	
${}_{i}$	ethanol	2.1847 × 10^14^	3.0433 × 10^1^	−7.7889 × 10^4^	0.9956	5.45	This work
${}_{j}$	water	1.2170 × 10^11^	−5.4992 × 10^1^	−5.6995 × 10^4^	0.9968	2.29	
${}_{i}$	ethanol	5.7744 × 10^11^	−9.0935 × 10^1^	−5.6031 × 10^4^	0.9971	2.51	[60]
${}_{j}$	water	1.5557 × 10^10^	8.7905 × 10^1^	−5.9686 × 10^4^	0.9947	2.76	
${}_{i}$	ethanol	4.7114 × 10^10^	3.3738 × 10^1^	−5.7050 × 10^4^	0.9773	6.54	[59]
${}_{j}$	water	1.4105 × 10^14^	−2.8712 × 10^2^	−5.9628 × 10^4^	0.9781	8.17	
${}_{i}$	ethanol	4.9447 × 10^6^	−9.1482 × 10^1^	−2.6643 × 10^4^	0.9869	4.56	[66]
${}_{j}$	water	9.8762 × 10^4^	−3.4970 × 10^1^	−2.0040 × 10^4^	0.9869	3.27	
${}_{i}$	ethanol	8.1045 × 10^8^	2.5624 × 10^−1^	−4.2036 × 10^4^	0.9921	4.58	[79]
${}_{j}$	water	4.6406 × 10^6^	−6.8452 × 10^1^	−2.6003 × 10^4^	0.9800	4.84	
${}_{i}$	ethanol	1.8201 × 10^11^	−3.6583	−5.2039 × 10^4^	0.9896	6.91	[80]
${}_{j}$	water	1.7481 × 10^10^	−5.8910 × 10^1^	−4.5465 × 10^4^	0.9935	4.34	
${}_{i}$	water	2.3440 × 10^13^	−3.4132 × 10^1^	−6.8774 × 10^4^	0.9677	9.37	[81]
${}_{j}$	ethanol	8.8538 × 10^20^	−2.6617	−1.3201 × 10^5^	0.9939	7.23	
${}_{i}$	water	5.5648 × 10^6^	1.6967 × 10^1^	−3.0690 × 10^4^	0.9917	4.70	[45]
${}_{j}$	1-butanol	1.3693 × 10^9^	−4.8434 × 10^2^	−1.6507 × 10^4^	0.9923	7.52	

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
