# Peer review of "Separation and Semi-Empiric Modeling of Ethanol–Water Solutions by Pervaporation Using PDMS Membrane"

_polymers, 2020, doi:10.3390/polym13010093_

Round 1

Reviewer 1 Report

The article entitled “Separation and semi-empiric modeling of ethanol-water solutions by pervaporation using PDMS membrane” is an original research articles which present good insights in the field of hydrophobic pervaporation. The use of hydrophobic pervaporation is proposed as a candidate for the direct separation of ethanol in diluted aqueous systems. The authors have properly characterized the pervaporation performance of the commercial PDMS membrane. The article can be accepted for publication after attending the minor comments: 1) Intro: The authors should also mention that hydrophobic PV is currently use for the extraction of aromas from agro-food systems (https://doi.org/10.1016/j.tifs.2019.12.003), and ethanol removal from alcoholic drinks towards the production of non-alcoholic beverages (https://doi.org/10.1080/07388551.2019.1631248 ). Please, check and cite the mentioned reported which will support your feedback. 2) Experimental: Over the experimental part, all experiments and modeling part has been well described. 3) Results: No comment at all, they have fully described and discussed their results. The authors have shown clear trend in PV performance as a function of temperature and feed concentration. In fact, they have properly discussed their permeation data using Arrhenius relationship, confirming a temperature dependency of the molecule’s permeation. In addition, the authors have compared their results with the current literature. Here, I would like to suggest to the authors to point out the importance of reporting the data in terms of permeance to provide a fair comparison among studies. “According to the report of Baker et al. (2010), permeance (Πi) is a better way of reporting pervaporation results when a fair comparison of different studies is needed. The permeance is a parameter independent from the driving force, allowing to compare experiments at different feed concentrations, feed temperatures and permeate pressures”. The authors may give a similar feedback in their article. 4) Conclusions: More than provide the concluding remarks of the study, the authors may also provide the research gap of hydrophobic pervaporation, as well as new recommendations for the new researchers in the field.

Reviewer 2 Report

Please find the comments in attached pdf file.

Reviewer 3 Report

Authors report on the semi-empirical model of pervaporation transport through PDMS membrane. The subject is interesting and fits the profile of the journal; however manuscript must be substantially revised before publishing, according to the following comments/suggestions.
1. Abstract should be improved, especially lines 10-14.
2. Introduction – should be extended, by adding the short overview of the approaches available in the literature.
3. Line 42 “latter” not “last”
4. Section 2.3 – more information regarding the used membrane is needed, i.e. producer (as it is a commercial product), membrane area, length of the tube.
5. Section 2.4 – the description should be improved, as it is not clear why the stabilization was needed without vacuum? What about reaching the stationary state during pervaporation experiments (i.e. building the concentration profile across membrane)? How many times the experiments were repeated? Eq. 4 is wrong – the symbol “beta” (β) must be used instead of “alpha” (α), according to the metrics introduced by R. Baker et al. (J. Membrane Sci. 2010). Moreover, there is one coefficient, i.e. “enrichment factor” called by Authors as “increased concentration” (Fig. 2b) or in a more a convoluted explanation in lines 100-101 “it is recommended to analyze the separation factor based on the number of times that the components on the feed are concentrated in the permeate, according to Equation (4)”. The citing of Eq (4) is not proper. Enrichment factor EF is qual to the ratio of the ethanol content in permeate to the ethanol content in the feed (see the paper by Baker et al.).
6. Section 2.5 – more detailed description is needed, algorithm of the calculations. There are readers who are not familiar with OriginLab software.
7. Results – general comment – practically all figures should be redrawn, by adding the error bars as well as legend. It is difficult to analyze the plot looking for the explanation of symbols in the figure caption.
8. Fig. 2 – an information regarding the separation efficiency is given 3 times. This is not necessary – Authors must decide which is the most important (i.e. separation factor, enrichment factor, or ethanol content in the permeate). Moreover, the total fluxes are much less informative than partial fluxes. The additional plot would be interesting, i.e. comparison of PV with vapor liquid equilibrium (distillation).
9. Lines 110-113 – in fact, the most important is an increase in the driving force and that should be clearly stated.
10. Fig. 4 – please check the description of x axes. The same scale on y axes would allow for the better comparison of the results.
11. Fig. 5 – do not connect the experimental points by the segmented lines (dashed line). The experimental points should be marked with the error bars and then the linear trend can be added.
12. Eq. (8) – I cannot agree with the proposed notation of this equation. It is true for the flux of ethanol, but not in the case of water flux. At ethanol concentration equal to 0 (pure water in feed) the permeate flux is non-zero (as it can be seen in Fig. 3b).
13. Tab. 2 – please double check if the feed composition refers to ethanol or to water (especially last raw, as the Pervap 2510 is a hydrophilic membrane), other errors: raw 1, temperature range of your results, header – separation factor must be “beta” not “alpha”
14. Conclusions – must be strengthen. You should also compare the results with other approaches (models).
15. Lines 191-197 – please refer to the information for Authors with regards to the information which should be provided as “Funding” and “Acknowledgements”.
16. Abbreviations – please correct language errors, e.g. t – time (not “tempo”), “separation factor” not “factor separation”; “i”, “j” – it is not “Permeate component”; I suggest to put “i” as the mixture component (either feed or permeate).
17. Appendix A.1 – some comments should be added to each Figure, related to the system (membrane, process conditions). Although these results are gathered in Table 2, this is insufficient.

Final recommendation – reconsider after the major revisions.

Round 2

Reviewer 3 Report

Authors improved the text, however in the revised version all the changes should be clearly marked, e.g. using red font or the changed text should be the part of the "Responses to reviewer comments" file.